# Identification, Characterization, and Homology Analysis of a Novel Strain of the Crimean–Congo Hemorrhagic Fever Virus from Yunnan, China

**DOI:** 10.3390/microorganisms12071466

**Published:** 2024-07-19

**Authors:** Jiale Wang, Taif Shah, Jiuxuan Zhou, Xinhua Long, Yixuan Wang, Jie Chen, Mingfei Shi, Zahir Shah, Binghui Wang, Xueshan Xia

**Affiliations:** 1Faculty of Life Science and Technology, Kunming University of Science and Technology, Kunming 650500, Chinataifshah@yahoo.com (T.S.);; 2State Key Laboratory for Conservation and Utilization of Bio-Resources in Yunnan, Yunnan University, Chenggong, Kunming 650091, China; 3Research Institute of Forest Protection, Yunnan Academy of Forestry and Grassland, Kunming 650500, China; 4Yunnan Province Baimaxueshan National Nature Reserve Management Bureau, Shangri-La 674400, China; 5College of Veterinary Sciences, The University of Agriculture, Peshawar 25130, Pakistan; drzahir@aup.edu.pk; 6School of Public Health, Kunming Medical University, Kunming 650500, China

**Keywords:** *Capricornis milneedwardsii*, Crimean–Congo hemorrhagic fever virus, characterization, homology, Yunnan

## Abstract

Wildlife serve as potential microbial reservoirs, accounting for approximately 70% of emerging infectious diseases. Crimean–Congo hemorrhagic fever virus (CCHFV), which causes Crimean–Congo hemorrhagic fever (CCHF) in humans, is a highly pathogenic tick-borne virus prevalent in several parts of Asia, Africa, and Europe with high case fatality rates. Several CCHFV cases have been reported in Asia, the Middle East, Africa, and Southern and Eastern Europe. The disease is endemic in several parts of western China, particularly Xinjiang. Ticks of the genus *Hyalomma* have been identified as a principal vector and reservoir for CCHFV, although other tick species may also have a crucial role in maintaining CCHFV in endemic regions. On infection, CCHF begins as a nonspecific febrile illness that can progress to severe hemorrhagic manifestations with a higher case fatality due to the unavailability of vaccines or other therapeutic agents. In this study, we collected tissue samples from a wild dead Chinese serow (*Capricornis milneedwardsii*) and three *Naemorhedus griseuses* from Deqin County, Tibetan Autonomous Prefecture, Yunnan, China, to investigate for contagious viruses that could be transmitted to humans. We identified a novel CCHFV strain, YNDQL-415G, in the liver tissue of a dead *C. milneedwardsii*. We performed nucleotide and amino acid sequence homology on the full-length viral genome. The results revealed significant homology between the viral S segment to that of the Africa1 strain, while the M and L segments showed similarity with the Asia CCHFV strain, indicating potential gene reassortment in the YNDQL-415G strain. The genetic characterization of a novel CCHFV strain from a dead *C. milneedwardsii* raises concerns about the possibility of a new zoonotic infection. A regular survey program is recommended to track the distribution of wild animals as well as the viruses they may transmit to humans and other domestic mammals in the region.

## 1. Introduction

The Crimean–Congo hemorrhagic fever virus (CCHFV) was first identified as a cause of febrile illnesses in the Congo [1]. Further investigation of febrile illnesses in Crimea revealed that the virus-causing cases were antigenically identical to the illnesses in the Congo [2]. Human studies have shown that CCHFV is a widely distributed virus in Southeast Asia, the Middle East, Africa, and Southern and Eastern Europe [2]. Ticks of the genus *Hyalomma* are primarily reservoirs and vectors of CCHFV, though other ticks may also keep the virus alive in endemic areas [3]. 

Transmission of CCHFV-infected ticks on migratory birds and global transportation may result in tick dispersal to new regions, such as the tick *Hyalomma,* to Sweden [4]. Serological studies have shown that CCHFV can infect a variety of wild animals, including hares, buffalo rhinoceroses, rodents, and ostriches [5]. Although most infected mammals initially show no obvious disease symptoms, CCHFV-infected animals are potential hosts, enabling the virus to spread from infected ticks to healthy ones after feeding on their blood [6]. CCHFV infects humans through tick bites, handling or butchering of infected livestock, or contact with virus-contaminated surfaces. On infection, CCHF begins as a nonspecific febrile illness, such as high fever, vomiting, diarrhea, and muscle aches, that can lead to severe hemorrhage, multi-organ failure, or death in severe cases [1]. Case fatality rates are higher than 30% in some regions due to the unavailability of approved vaccines or other therapeutic agents. 

CCHFV is an enveloped negative-sense RNA virus belonging to the genus *Orthonairovirus* in the Nairoviridae family [1]. The viral genome is composed of three gene segments: a small (S) segment that encodes the nucleocapsid protein (NP), a medium (M) segment that encodes the glycoprotein (G) precursor, and a large (L) segment that encodes RNA-dependent RNA polymerase (RdRp) [1]. As CCHFV is a segmented RNA virus, its genomic rearrangements play a significant role in its evolution and geographic distribution. The M segment has undergone more frequent rearrangements than the S or L segments [7]. The S segment of CCHFV has been used for phylogenetic analysis and revealed eight distinct subtypes associated with specific geographic regions [8], specifically two in Asia, three in Europe, and three in Africa. 

Yunnan is foremost among the Chinese provinces for hosting diverse animal species, with more than 250 species of mammals, 780 birds, rare animals (such as slow loris, snub-nosed monkeys, bison, and hornbill), and protected animals (such as *Capricornis milneedwardsii*, Assamese macaque, Phayre leaf monkey, musk deer, and red panda). *C. milneedwardsii* is under second-class national protection and can leap over rocks and run fast in forests and mountainous areas. The *C. milneedwardsii* is widely distributed in the Southwest and Southeast China regions, including the area from southern Gansu southward through Sichuan and most of the Yunnan regions. 

Over the past few years, there has been a significant increase in re-emerging infectious diseases transmitted across species. These diseases pose a major threat to public health and have caused significant harm to both human health and public property. Wild animals gain importance in transmitting zoonotic diseases, as they serve as reservoirs for several zoonotic viral pathogens [9]. Therefore, disease surveillance in wildlife can provide crucial insights into pathogens before they spread widely. This proactive approach allows for advanced preventive measures and the establishment of an effective surveillance system to protect humans and animals from future zoonotic outbreaks. 

In this study, we identified, characterized, and phylogenetically analyzed a novel CCHFV strain named YNDQL-415G from the liver tissue of a dead *C. milneedwardsii* in Deqin County, Tibetan Autonomous Prefecture, Yunnan, China. We performed nucleotide and amino acid homology analyses of the full-length viral genome. The results revealed significant homology between the S segment and that of the Africa1 strain, while the M and L segments showed similarity with the Asia strain, suggesting potential gene reassignment in the CCHFV strain YNDQL-415G. 

## 2. Materials and Methods

In April 2021, the research team reported the unnatural deaths of a wild *C. milneedwardsii* and three *Naemorhedus griseuses* in the Bema Xueshan Benzilan Nature Reserve, Diqing Tibetan Autonomous Prefecture, Yunnan Province, China. Tissue samples, including liver, kidney, spleen, heart, small intestine, skin, and blood, were collected from each dead animal immediately with strict measures to prevent contamination. Ticks (n = 6) near and on their bodies were also collected and dispensed into fresh sample tubes. Each collected sample was stored in a virus preservation solution in the presence of PBS, transported on dry ice, and subsequently stored in a −80 °C freezer in the laboratory. Before the experiment, each sample was split, and a small amount of tissue was used for next-generation sequencing. 

### 2.1. Viral Genomic RNA Extraction and Sequencing

Viral genomic RNA was extracted from each sample using the TaKaRa MiniBEST Viral RNA/DNA Extraction Kit (Takara, Beijing, China). The NanoDrop One Spectrophotometer and Qubit 4 Fluorometer (Thermo Fisher Scientific, Waltham, MA, USA) were used to measure the concentration and quality of genomic RNA. Finally, the REPLI-g Cell WGA & WTA Kit (Qiagen, Hilden, Germany) was used for genome amplification, and the amplified PCR product was selected on a 1% agarose gel for library construction. Sequencing libraries were generated using the ALFA-SEQ DNA Library Prep Kit for Illumina (FINDROP, Guangzhou, China) following the manufacturer’s guidelines. The library quality was assessed on the Qubit dsDNA HS Assay Kit (Thermo Fisher Scientific, Waltham, MA, USA) before sequencing on an Illumina Novaseq 6000 system (Illumina San Diego, CA, USA). 

### 2.2. CCHFV Confirmation 

In order to confirm the presence of the CCHFV in the liver of a dead *C. milneedwardsii*, total genomic RNA was extracted from 200 µL of the infected tissue homogenates using a TIANamp virus RNA kit (TIANGEN, Beijing, China) in accordance with the manufacturer’s instructions. Complementary DNA (cDNA) was synthesized from the extracted RNA and then subjected to PCR amplification with a gene-specific primer set.

As the CCHFV genome comprises three segments, namely large (L), medium (M), and small (S), for the S segment amplification, which was not screened by the Illumina platform, we performed nested PCR with segment-specific primers (peripheral primer F1: TGGACACTTTCACAAACTC, R1: GACAAATTCCCTGCACCA) for the first step of the PCR reaction, which amplified a 540 bp segment. We used another primer set (F2: GAGTGTGCCTGGTTAGTTC, R2: GACATTACAATTTCACCAGG) for the 260-bp segment amplification. The conditions set for the PCR amplification were as follows: initial denaturation at 94 °C for 5 min, followed by 36 cycles (denaturation at 94 °C for 30 s, annealing at 50 °C for 30 s, and extension at 72 °C for 40 s). The final extension was performed at 72 °C for 7 min [10]. For virus confirmation, the PCR-amplified product was Sanger sequenced, and the nucleotide sequences were NCBI-blast.

## 3. Results and Discussion

In our findings, only the live tissue of a dead *C. milneedwardsii* was found to contain CCHFV out of all the collected samples from animals and six whole ticks from Bema Xueshan Benzilan Nature Reserve, Diqing Tibetan Autonomous Prefecture, Yunnan. The three genomic segments, large (L), medium (M), and small (S), of CCHFV were confirmed with nested PCR with a segment-specific primer set. The three discontinuous lines in the genome map’s outermost circle represent the L, M, and S segments. 0 kb represents the start site of the gene (ATG), and the L segment is labeled at the 10 kb position, and those less than 10 kb are not labeled (Appendix A).

We further performed amino acid and nucleotide sequence homology analyses of the three segments, L (11.870 kb), M (5.134 kb), and S (1.48 kb), of the newly identified CCHFV strain with other representative viral strains in the NCBI database (Figure 1). Each column in the table represents nucleotide sequence homology, followed by amino acids. Our finding showed that the L segment, our YNDQ-L415GL strain, showed the highest (82.23%) nucleotide sequence identity with a previously identified Asia 1 strain, followed by China_2020 (79.95%) and Africa_1 (79.95%). Similarly, based on the amino acid, the YNDQ-L415GL strain showed the highest homology with Africa_2 (92.78%), followed by China_2020 (92.60%) and Asia_1 (92.50%).

Based on the M segment, our YNDQ-L415GL strain showed the highest nucleotide sequence identity (80.67%) with a previously identified Asia_3 strain, followed by Africa_2 (79.66%) and Africa_1 (79.63%). Similarly, based on the amino acid, the YNDQ-L415GL strain showed the highest homology with Asia_3 (99.12%), followed by Africa_2 (98.77%) and Europe_2 (92.50%). The virus is suspected of producing adaptive mutations in the M segment after infecting a new host, resulting in decreased nucleotide similarity. 

According to the S segment, our YNDQ-L415GL strain showed the highest nucleotide sequence identity (87.03%) with a previously identified Africa 1 strain, followed by China_2020 (84.47%) and Asia_1 (84.27%). Similarly, based on the amino acid sequence, the YNDQ-L415GL strain showed the highest homology with Africa_1 (96.07%), followed by Europe_1 (92.75%), and China_2020 (92.75%). Therefore, we are more skeptical that such segmented viruses are more susceptible to genomic rearrangements and potentially more harmful to hosts. 

To investigate the full-length gene sequence of L, M, and S segments of the CCHFV strain YNDQ-L415GL, we performed phylogenetic tree analysis (Figure 2) to understand viral genomic homology. According to the analysis, the S segment exhibited the highest nucleotide sequence homology (87.03%) with CCHFV (DQ211640.1), isolated from goats in Senegal, belonging to the African type. The M segment exhibited a nucleotide sequence identity (81.7%) with CCHFV (MH688498.1) isolated from *Hyalomma asiaticum* in China, belonging to the Asian type. On the other hand, the L segment showed a nucleotide sequence identity (80.58%) with CCHFV (MN930405.1) isolated from humans in India, belonging to the European type 2. Currently, CCHFV is genotyped based on the conserved S segment.

According to the available literature, CCHFV is widespread in the northwestern regions of Xinjiang, China, and Inner Mongolia [11]. The predominant strain in these areas is Asian type 2, while the S segment of CCHFV found in this study was most similar to the conserved African type 1 strain. The African type 1 strain is primarily found in Africa, possibly as a result of genetic fragmentation caused by migratory birds carrying infected ticks and reassortment [12]. The study discovered that the migratory paths of birds in West and East Africa are similar to those in West and Central Africa and Northwest China. This strain differs from the strains previously prevalent in China [12,13]. The World Health Organization (WHO) has classified CCHF as a blueprint priority disease and a priority for research during public health emergencies. However, there is a lack of systematic publication of CCHFV data in Central, East, and Southeast Asia [14]. 

Yunnan Province is known for its rich animal resources, which facilitate close exchanges between wild species. Livestock farming in Yunnan Province typically involves free-range farming, which increases the likelihood of contact between livestock and wildlife. This contact can raise the risk of the transmission of viruses from animals to humans. Studies have shown that individuals close to livestock are at an elevated risk of contracting CCHF, potentially because livestock act as intermediate hosts for transmission [15]. Additionally, its proximity to several Southeast Asian countries has led to a strong emphasis on border trade and domestic and foreign tourism. As a result, Yunnan has become a hub for cooperation and exchange between China and Southeast Asia and is also an essential region for public health and safety [16]. The first detection of CCHFV in *C. milneedwardsii* marks a significant breakthrough for molecular epidemiological research on the virus in both Yunnan and Southeast Asia. As one of the causes of critical zoonotic diseases, this discovery also offers a valuable reference for future research focusing on the virus’s potential expansion of host range and the emergence of new genotypes [15]. To determine if the virus had spread to other wildlife, we conducted tests on rodents (particularly rats and squirrels), insectivores (odorous shrews), and mammals (cattle and sheep) in the region. However, none of the tested animals were positive for the presence of CCHFV. Similarly, screening of six tick samples collected on or around the dead animal did not reveal the presence of CCHFV, indicating that animal species act as crucial CCHFV amplifying hosts, facilitating the virus’s transmission from infected ticks to uninfected hosts. 

## Figures and Tables

**Figure 1 microorganisms-12-01466-f001:**
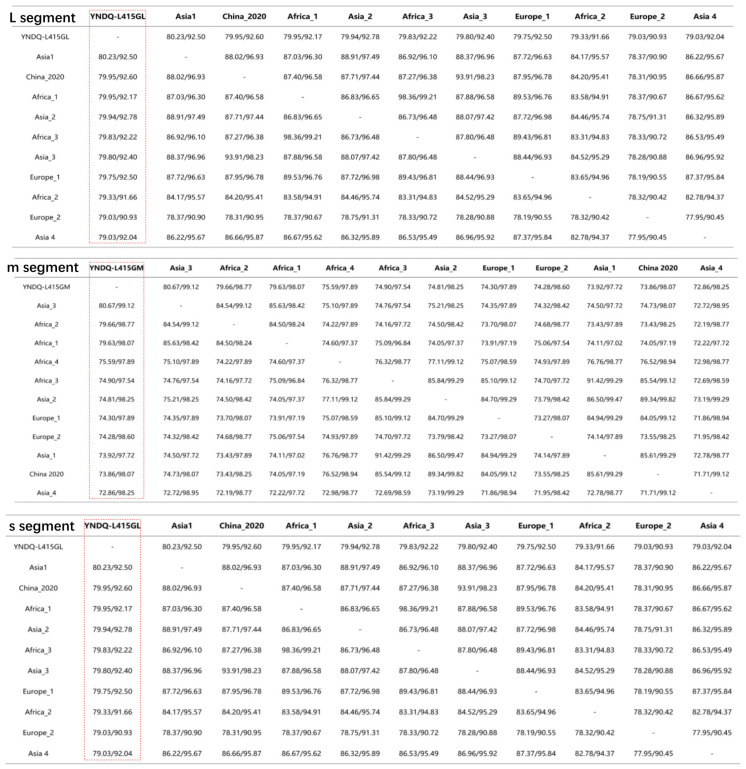
Nucleotide and amino acid sequence homology analysis of our CCHFV strain S segment (1.48 kb), M segment (5.134 kb), and L segment (11.870 kb) with some of the previously reported viral subtypes using BioAider software (v.1.532).

**Figure 2 microorganisms-12-01466-f002:**
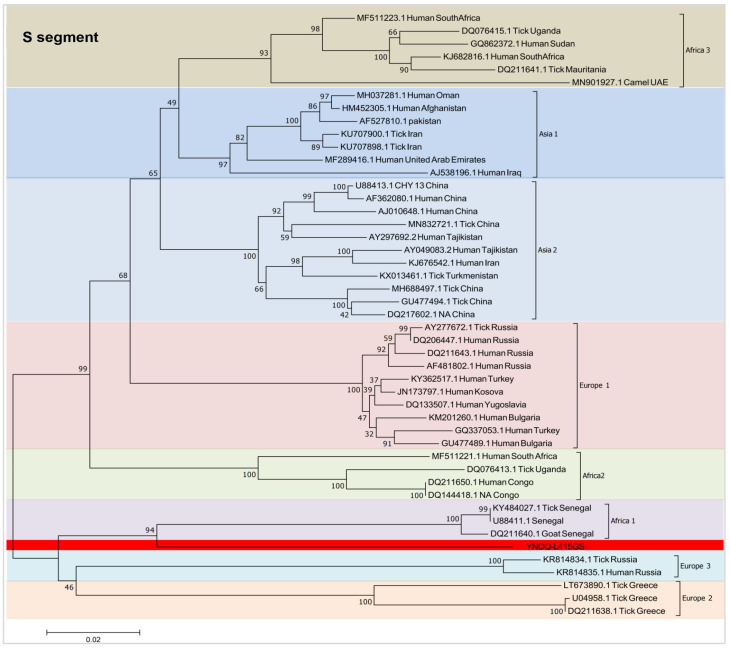
Maximum likelihood tree analysis using the complete CCHFV genome sequence obtained from NCBI and the sequences from this study. Sequences of complete segments were aligned using ClustalW program in Mega 7.0 and analyzed for maximum likelihood using the GTR+G+I model with a bootstrap value of 1000. Branches are labeled with the NCBI accession number, source of organism, and origin country. The term “NA” indicates that no information was found. Sequences from this study are shown in red font. S segment (1.480 kb), M segment (5.134 kb), and L segment (11.870 kb).

## Data Availability

The CDS regions of the L, M, and S segments have been submitted to the NCBI database under the accession numbers (L: OQ633002, S: OQ633004, and M: OQ633003).

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
