# Peer review of "Identification, Characterization, and Homology Analysis of a Novel Strain of the Crimean–Congo Hemorrhagic Fever Virus from Yunnan, China"

_microorganisms, 2024, doi:10.3390/microorganisms12071466_

Round 1
Reviewer 1 Report
Comments and Suggestions for Authors
The manuscript provides data on the isolation of a new strain of CCHFV, YNDQL-415G, in dead Chinese sero (Capricornis milneedwardsii), Tibet Autonomous Prefecture, Yunnan, China. The topic is extremely interesting, because in the coming years new strains of pathogenic viruses are expected to appear in the human population, which are the result of a process of reassortment and zoonotic infection transferred to humans. And any such possibility should be analyzed in advance so that appropriate prevention strategies can be devised. I consider that the authors have detailed the methods they used, the results and, accordingly, future directions for further analyses.The supplementary materials are clearly presented. The references in the bibliography are mostly from the last 5 years, which meet the requirements.
Author Response
Dear Editor,
We highly appreciate your efforts and consideration of our manuscripts' as a possible publication. Please convey our hearty thanks to the referees for their constructive and positive comments and suggestions. We have revised this manuscript entitled "Identification, characterization, and homology analysis of a novel strain of the Crimean-Congo hemorrhagic fever virus from Yunnan, China" under manuscript ID: "microorganisms-3075339" carefully according to their suggestions. However, if something is missing, we will gladly revise the manuscript. The details of the comments and their answers are given below.
Author's reply to the Reviewer 1
- The manuscript provides data on the isolation of a new strain of CCHFV, YNDQL-415G, in dead Chinese serow (Capricornis milneedwardsii), Tibet Autonomous Prefecture, Yunnan, China. The topic is extremely interesting, because in the coming years new strains of pathogenic viruses are expected to appear in the human population, which are the result of a process of reassortment and zoonotic infection transferred to humans. And any such possibility should be analyzed in advance so that appropriate prevention strategies can be devised. I consider that the authors have detailed the methods they used, the results and, accordingly, future directions for further analyses. The supplementary materials are clearly presented. The references in the bibliography are mostly from the last 5 years, which meet the requirements.
Response: Thank you very much, we are very grateful for the reviewer’s appreciation.
We hope these justifications are acceptable to the editorial board; however, if there is any further concern, we will revise according to the suggestions.
We once again thank you so much for holding our manuscript for publication.
Sincerely,
Prof. Xueshan Xia
Faculty of Life Science and Technology,
Kunming University of Science and Technology, Kunming, 650500, Yunnan, P.R. China & Yunnan Provincial Center for Molecular Medicine
E-mail: oliverxia2000@aliyun.com
Reviewer 2 Report
Comments and Suggestions for Authors
In this submission, authors did identification, characterization and genomic analysis of Crimean-Congo Hemorrhagic fever virus (CCHFV) from Yunnan, China. I have several comments;
What is the objective and aims of this study?
Please follow the citation rule for the reference: '..s in the Nairoviridae family (Hawman and Feldmann, 2023)...' (Line 61).
Please explain about Capricornis milneedwardsii and the objective in relevance to collecting samples from this species in the introduction.
Where did these samples come from: Capricornis milneedwardsii and three Naemorhedus griseuses? Were they found in natural parks, or were they captive animals? Did a hunter find them dead? How and by whom were the researchers notified about these animals? How did they come across these animals? How many animals were found, where, and at what time? When were samples collected after they were found dead?
The Material and Methods section is briefly written and is not sufficient. Please make subtitles for each method being used with specific catalog numbers of the reagents used. Please also address the methods used for the data analysis.
What are the results per sample manner? Which samples showed positivity?
What is '.....the tested animals wad 201 positive for the presence of CCHFV'? at line 201. Is there a typo in 'wad'?
What do the authors mean by 'Similarly, screening of samples collected around the dead animal did not reveal the presence of CCHFV, indicating that animal species act as crucial CCHFV amplifying hosts, facilitating the virus's transmission from infected ticks to uninfected hosts'? What are those animals that have been tested for the presence of CCHFV? Please indicate the species of other animals, their number, and how sampling was made. Were they killed for sampling, or was blood just collected? Please indicate all in the methods.
Author Response
Dear Editor,
We highly appreciate your efforts and consideration of our manuscripts' as a possible publication. Please convey our hearty thanks to the referees for their constructive and positive comments and suggestions. We have revised this manuscript entitled "Identification, characterization, and homology analysis of a novel strain of the Crimean-Congo hemorrhagic fever virus from Yunnan, China" under manuscript ID: "microorganisms-3075339" carefully according to their suggestions. However, if something is missing, we will gladly revise the manuscript. The details of the comments and their answers are given below.
Author's reply to Reviewer 2
In this submission, authors did identification, characterization and genomic analysis of Crimean-Congo Hemorrhagic fever virus (CCHFV) from Yunnan, China. I have several comments;
- What is the objective and aims of this study?
Response: Wildlife serve as potential microbial reservoirs, accounting for approximately 70% of emerging infectious diseases. In this study, we collected tissue samples from a wild dead Chinese serow (Capricornis milneed-wardsii) and three Naemorhedus griseuses from Deqin County, Tibetan Autonomous Prefecture, Yunnan, China, to investigate for contagious viruses that could be transmitted to humans.
- Please follow the citation rule for the reference: '..s in the Nairoviridae family (Hawman and Feldmann, 2023)...' (Line 61).
Response: Thank you very much, we have revised the citation, i.e., …genus Orthonairovirus in the Nairoviridae family [1].
- Please explain about Capricornis milneedwardsii and the objective in relevance to collecting samples from this species in the introduction.
Response: Thank you very much, we have revised and added, “Yunnan is foremost among the Chinese provinces for hosting diverse animal species, with more than 250 species of mammals, 780 birds, rare animals (such as slow loris, snub-nosed monkeys, bison, and hornbill), and protected animals (such as Capricornis milneedwardsii, Assamese macaque, Phayre leaf monkey, musk deer, and red panda). C. milneedwardsii is under second-class national protection and can leap over rocks and run fast in forests and mountainous areas. The C. milneedwardsii is widely distributed in the Southwest and southeast China regions, including the area from southern Gansu south-ward through Sichuan and most of the Yunnan regions”.
- Where did these samples come from: Capricornis milneedwardsii and three Naemorhedus griseuses? Were they found in natural parks, or were they captive animals? Did a hunter find them dead? How and by whom were the researchers notified about these animals? How did they come across these animals? How many animals were found, where, and at what time? When were samples collected after they were found dead?
Response: Thank you very much, we have revised and added, “In April 2021, the research team reported the unnatural deaths of a wild C. milneed-wardsii and three Naemorhedus griseuses in the Bema Xueshan Benzilan Nature Reserve, Diqing Tibetan Autonomous Prefecture, Yunnan Province, China. Tissue samples, in-cluding liver, kidney, spleen, heart, small intestine, skin, and blood, were collected from each dead animal immediately with strict measures to prevent contamination. Ticks (n = 6) near and on their bodies were also collected and dispensed into fresh sample tubes. Each collected sample was stored in a virus preservation solution in the presence of PBS, transported on dry ice, and subsequently stored in a -80°C freezer in the laboratory. Before the experiment, each sample was split, and a small amount of tissue was used for next-generation sequencing”.
- The Material and Methods section is briefly written and is not sufficient. Please make subtitles for each method being used with specific catalog numbers of the reagents used. Please also address the methods used for the data analysis.
Response: Thank you very much, we have revised MM section carefully, however, according to the journal rule, communication can be limited to 2000 words, and still our article text is around 24000 words. So, we cannot add additional text, according to the journal requirement.
- What are the results per sample manner? Which samples showed positivity?
Response: At the beginning of result section, we have added “In our findings, only the live tissue of a dead C. milneedwardsii was found to contain CCHFV out of all the collected samples from animals and six whole ticks from Bema Xueshan Benzilan Nature Reserve, Diqing Tibetan Autonomous Prefecture, Yunnan. The three genomic segments, large (L), medium (M), and small (S), of CCHFV were confirmed with nested PCR with a segment-specific primer set”.
- What is '...the tested animals wad 201 positive for the presence of CCHFV'? at line 201. Is there a typo in 'wad'?
Response: We have checked this line for correction.
- What do the authors mean by 'Similarly, screening of samples collected around the dead animal did not reveal the presence of CCHFV, indicating that animal species act as crucial CCHFV amplifying hosts, facilitating the virus's transmission from infected ticks to uninfected hosts'? What are those animals that have been tested for the presence of CCHFV? Please indicate the species of other animals, their number, and how sampling was made. Were they killed for sampling, or was blood just collected? Please indicate all in the methods.
Response: Ticks (n = 6) near and on their bodies were collected and dispensed into fresh sample tubes. Each collected sample was stored in a virus preservation solution in the presence of PBS, transported on dry ice, and subsequently stored in a -80°C freezer in the laboratory.
We hope these justifications are acceptable to the editorial board; however, if there is any further concern, we will revise according to the suggestions.
We once again thank you so much for holding our manuscript for publication.
Sincerely,
Prof. Xueshan Xia
Faculty of Life Science and Technology,
Kunming University of Science and Technology, Kunming, 650500, Yunnan, P.R. China & Yunnan Provincial Center for Molecular Medicine
E-mail: oliverxia2000@aliyun.com
Reviewer 3 Report
Comments and Suggestions for Authors
Manuscript ID: microorganisms-3075339 - Review for Microorganisms Date:28/06/2024
Title
Identification, Characterization, and Homology Analysis of a 2 Novel Strain of the Crimean-Congo Hemorrhagic Fever Virus 3 From Yunnan, China
Abstract: The abstract is informative and bring up the main points of the article, with a due introduction to the matter and a brief description of the disease, essentially pointing out to the possibility of occurrence of potential gene reassortments in the virus.
Introduction: The introduction starts with a brief comment of the epidemiology of CCHF and the mains characteristics of CCHFV, then heading towards the main objective of the paper.
Mat and Met:
In the description of the case, line 93, instead of “.... Yunnan Province, discovered the unnatural death...” I suggest to substitute “discovered” by “reported”.
In line100-101: Co extraction of... you mean extraction of total DNA and RNA? PLease review the sentence.
Line 102: “...and the whole genome was amplified with Qiagen kit...” with what kind of primers? Please specify.
The sentence in lines 100 to 103 is too long. Use points to separate subjects.
Results and discussion
Results and discussion are presented together and focus on the main finding of the article, which is the sequence of the new CCHFV recovered, while discussing the relevant points in the findings reported, comparing the sequence with Other similar sequences available in the literature and genomic databases. The figures are informative and well presented,
They conclude article stating that “The first detection of CCHFV in a Chinese serow (Capricornis milneedwardsii) marks a significant breakthrough for molecular epidemiological research on the virus in both Yunnan and Southeast Asia. As one of the causes of critical zoonotic diseases, this discovery also offers a valuable reference for future research focusing on the virus's potential expansion of host range and the emergence of new genotypes.”, with which I totally agree.
Summarizing my review, I have nothing else to add beyond the comments I made above. I suggest that the authors have a look at my comments and make the due alterations if they think those are appropriate. I believe the paper is nearly ready for publication. Congrats to the authors.
Comments on the Quality of English LanguageThe written English is quite clear, although in a few sentences I took the liberty of suggesting some minor alterations to improve the "fluidity" of the text.
Author Response
Dear Editor,
We highly appreciate your efforts and consideration of our manuscripts' as a possible publication. Please convey our hearty thanks to the referees for their constructive and positive comments and suggestions. We have revised this manuscript entitled "Identification, characterization, and homology analysis of a novel strain of the Crimean-Congo hemorrhagic fever virus from Yunnan, China" under manuscript ID: "microorganisms-3075339" carefully according to their suggestions. However, if something is missing, we will gladly revise the manuscript. The details of the comments and their answers are given below.
Author's reply to Reviewer 3
Title: Identification, Characterization, and Homology Analysis of a Novel Strain of the Crimean-Congo Hemorrhagic Fever Virus From Yunnan, China
Abstract: The abstract is informative and bring up the main points of the article, with a due introduction to the matter and a brief description of the disease, essentially pointing out to the possibility of occurrence of potential gene reassortments in the virus.
Introduction: The introduction starts with a brief comment of the epidemiology of CCHF and the mains characteristics of CCHFV, then heading towards the main objective of the paper.
Mat and Met: In the description of the case, line 93, instead of “.... Yunnan Province, discovered the unnatural death...” I suggest to substitute “discovered” by “reported”.
Response: we have revised and added, “In April 2021, the research team reported the unnatural deaths of a wild Capricornis milneedwardsii and three Naemorhedus griseuses in the Bema Xueshan Benzilan Nature Reserve, Diqing Tibetan Autonomous Prefecture, Yunnan Province, China”.
- In line100-101: Co extraction of... you mean extraction of total DNA and RNA? PLease review the sentence.
Response: Viral genomic RNA was extracted from each sample using the TaKaRa MiniBEST Viral RNA/DNA Extraction Kit (Takara, Beijing, China).
- Line 102: “...and the whole genome was amplified with Qiagen kit...” with what kind of primers? Please specify.
Response: Thank you very much; we have revised and added, “Finally, the REPLI-g Cell WGA & WTA Kit (Qiagen, Hilden, Germany) was used for genome amplification…”
- The sentence in lines 100 to 103 is too long. Use points to separate subjects.
Response: Thank you very much; we have revised the sentence.
Results and discussion
Results and discussion are presented together and focus on the main finding of the article, which is the sequence of the new CCHFV recovered, while discussing the relevant points in the findings reported, comparing the sequence with Other similar sequences available in the literature and genomic databases. The figures are informative and well presented,
They conclude article stating that “The first detection of CCHFV in a Chinese serow (Capricornis milneedwardsii) marks a significant breakthrough for molecular epidemiological research on the virus in both Yunnan and Southeast Asia. As one of the causes of critical zoonotic diseases, this discovery also offers a valuable reference for future research focusing on the virus's potential expansion of host range and the emergence of new genotypes.”, with which I totally agree.
Summarizing my review, I have nothing else to add beyond the comments I made above. I suggest that the authors have a look at my comments and make the due alterations if they think those are appropriate. I believe the paper is nearly ready for publication. Congrats to the authors.
Comments on the Quality of English Language
The written English is quite clear, although in a few sentences I took the liberty of suggesting some minor alterations to improve the "fluidity" of the text.
Response: Thank you very much, we have carefully revised the whole article, according to the reviewer’s suggestions.
We hope these justifications are acceptable to the editorial board; however, if there is any further concern, we will revise according to the suggestions.
We once again thank you so much for holding our manuscript for publication.
Sincerely,
Prof. Xueshan Xia
Faculty of Life Science and Technology,
Kunming University of Science and Technology, Kunming, 650500, Yunnan, P.R. China & Yunnan Provincial Center for Molecular Medicine
E-mail: oliverxia2000@aliyun.com
Round 2
Reviewer 2 Report
Comments and Suggestions for Authors
Thanks for addressing my comments.